This work is distributed under
the Creative Commons Attribution 4.0 License.

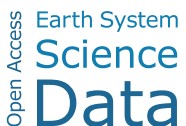

Earth System
Science
Data

# Reassessing the lithosphere: SeisDARE, an open-access seismic data repository

Irene DeFelipe[1], Juan Alcalde[1], Monika Ivandic[2], David Martí[1,a], Mario Ruiz[1], Ignacio Marzán[1], Jordi Diaz[1], Puy Ayarza[3], Imma Palomeras[3], Jose-Luis Fernandez-Turiel[1], Cecilia Molina[4], Isabel Bernal[4], Larry Brown[5], Roland Roberts[2], and Ramon Carbonell[1]

[1]Geosciences Barcelona, GEO3BCN, CSIC, c/ Lluís Solé i Sabarís, s/n, 08028, Barcelona, Spain
[2]Department of Earth Sciences, Uppsala University, 752 36, Uppsala, Sweden
[3]Department of Geology, University of Salamanca, Plaza de la Merced s/n, 37008, Salamanca, Spain
[4]Consejo Superior de Investigaciones Científicas, CSIC, c/Serrano, 117, 28006, Madrid, Spain
[5]Department of Earth and Atmospheric Sciences, Cornell University,
112 Hollister Drive, Ithaca, NY 14853-1504, USA
[a]now at: Lithica SCCL, Av. Farners 16, 17430, Sta. Coloma de Farners, Spain

**Correspondence:** Irene DeFelipe (irene.defelipe@gmail.com)

**Abstract.** CE1 Seismic reflection data (normal incidence and wide angle CE2) are unique assets for solid Earth sciences as they provide critical information about the physical properties and structure of the lithosphere as well as about the shallow subsurface for exploration purposes. The resolution of these seismic data is highly appreciated; however they are logistically complex and expensive to acquire, and their geographical coverage is limited. Therefore, it is essential to make the most of the data that have already been acquired. The collation and dissemination of seismic open-access data are then key to promote accurate and innovative research and to enhance new interpretations of legacy data. This work presents the Seismic DAta REpository (SeisDARE), which is, to our knowledge, one of the first comprehensive open-access online databases that stores seismic data registered with a permanent identifier (DOI). The datasets included here are openly accessible online and guarantee the FAIR (findable, accessible, interoperable, reusable) principles of data management, granting the inclusion of each dataset in a statistics referencing database so its impact can be measured. SeisDARE includes seismic data acquired in the last 4 CE3 decades in the Iberian Peninsula and Morocco. These areas have attracted the attention of international researchers in the fields of geology and geophysics due to the exceptional outcrops of the Variscan and Alpine orogens and wide foreland basins, the crustal structure of the offshore margins that resulted from a complex plate kinematic evolution, and the vast quantities of natural resources contained within. This database has been built thanks to a network of national and international institutions, promoting a multidisciplinary research and is open for international data exchange and collaborations. As part of this international collaboration, and as a model for inclusion of other global seismic datasets, SeisDARE also hosts seismic data acquired in Hardeman County, Texas (USA), within the COCORP project (Consortium for Continental Reflection Profiling). SeisDARE aims to make easily accessible old and recently acquired seismic data and to establish a framework for future seismic data management plans. SeisDARE is freely available at https://digital.csic.es/handle/10261/101879 (a detailed list of the datasets can be found in Table 1), bringing endless research and teaching opportunities to the scientific, industrial, and educational communities.

## 1   Introduction

Controlled-source seismology is a fundamental tool for solid Earth sciences, as it provides very valuable information about the physical properties and structure of the Earth's crust and upper mantle. Controlled-source seismic experiments have been undertaken in all tectonic settings since the mid-nineteenth century (Mintrop, 1947; Jacob et al., 1999), shedding light on our current understanding of the Earth's interior, from its composition and structure to its evolution. The outputs of these experiments, deep seismic sounding (DSS) data, are generally expensive, logistically complex to acquire, and often result from a huge scientific effort involving national and international institutions. Therefore, it is imperative that legacy and new acquired data are easily available for future generations of geoscientists. Taking into account that data-processing algorithms are constantly evolving, new or additional information can be extracted from these data.

Aware of the value of DSS data, Prodehl and Mooney (2012) provided a comprehensive, worldwide history of controlled-source seismological studies acquired from 1850 to 2005, highlighting the enormous efforts made in the last 170 years to increase our knowledge of the subsurface. Since the 1970s, ambitious seismic reflection projects have been developed worldwide, acquiring unique datasets that targeted key lithosphere structures. These datasets provided unprecedented images of the lithosphere, characterizing crustal discontinuities, major faults, geometry of orogenic belts, and other lithospheric features that had not been envisioned before. Some of these major seismic research programmes that acquired DSS normal incidence and wide-angle data include COCORP (USA); BIRPS (UK); ECORS (France); DEKORP (Germany); ESCI, IAM and ILIHA (Spain); HIRE and FIRE (Finland); ESRU and URSEIS (Russia); INDEPTH (Himalayas); LITHOPROBE (Canada); NFP 20 (Switzerland); and SINOPROBE (China).

In parallel, the hydrocarbon industry also added value using legacy seismic reflection data (Nicholls et al., 2015) by applying new processing approaches and techniques. Furthermore, reprocessed high-resolution (HR) seismic reflection data have also been used in exploration for mineral resources (Manzi et al., 2019; Donoso et al., 2020) and civil engineering (Martí et al., 2008) and are also successful for the characterization of seismogenic zones and hazard assessment (Ercoli et al., 2020). Therefore, legacy data are extremely valuable for basic and applied Earth sciences, and their preservation and availability constitute an effort that should contribute to move science forward.

The international community has focused on the preservation of the data, facilitating their access. Examples of these efforts include the SIGEOF database (Geological and Mining Institute of Spain) that hosts geophysical data of the Iberian Peninsula (http://info.igme.es/SIGEOF/, last access: March 2021), OpenFIRE (https://avaa.tdata.fi/web/fire, last access: March 2021, Finland), the IRIS-PASSCAL consortia (https://ds.iris.edu/ds/nodes/dmc/, last access: March 2021, USA), and the International Geological Correlation Programme (IGCP) initiative (http://www.earthscrust.org.au/, last access: March 2021).

Within the European research community there is a consistent vision of fostering global open science as a driver for enabling a new paradigm of transparent, data-driven science and accelerating innovation. This vision is becoming a reality thanks to a number of ambitious programmes internationally supported. The European Open Science Cloud (https://www.eosc-portal.eu/, last access: March 2021) is driving towards a virtual environment with open and seamless services for storage, management, analysis, and reuse of research data by federating existing scientific data infrastructures, currently dispersed across disciplines and European countries. In the same way, the European Strategy Forum on Research Infrastructures (ESFRI) is a strategic instrument to develop the scientific integration of Europe and to strengthen its international outreach. The European Plate Observation System (EPOS, https://www.epos-eu.org/, last access: March 2021), established under ESFRI, was initiated in 2002 to tackle a viable solution for solid Earth challenges. It is aligned with the Berlin Declaration for the Open Access to Knowledge in the Sciences and Humanities (2003) which stipulates that research data products need to be integrated and made accessible through open-access schemes. EPOS aims to ensure a long-term plan to facilitate the integration of data and fosters worldwide interoperability of Earth science services to a broad community of users for innovation in science, education, and industry.

In close collaboration with EPOS, the Spanish National Research Council (CSIC) hosts the "DIGITAL.CSIC" repository, where all kinds of scientific data are accessible following the international mandates of open-access data and the FAIR principles of data management: findable, accessible, interoperable, and reusable (Bernal, 2011; Wilkinson et al., 2016). As part of CSIC, Geosciences Barcelona (GEO3BCN, formerly Institute of Earth Sciences Jaume Almera – ICTJA) has participated in numerous geophysical projects carried out mainly in the Iberian Peninsula and Morocco during its more than 50 years' of history. The unique geological characteristics of these areas (outstanding record of the Variscan orogen and Alpine mountain belts, wide foreland basins, good accessibility and outcrop conditions, etc.) make them an excellent target to study the structure of the lithosphere and the spatial and temporal evolution of tectonic extension, mountain building, plate tectonics, and evolution of rifts (Daignières et al., 1981, 1982; Gallart et al., 1980, 1981; Choukroune and ECORS Team, 1989; Roca et al., 2011; Simancas et al., 2013; Martínez Poyatos et al., 2012; Macchiavelli et al., 2017; Ruiz et al., 2017; Cadenas et al., 2018; DeFelipe et al., 2018, 2019). These data have provided a full picture of the physical structure of the Mohorovičić discontinuity (Carbonell et al., 2014b; Díaz et al., 2016), the accommodation of shortening mechanisms at different crustal levels (Simancas

et al., 2003, 2013), the structure of the Iberian margins and mountain ranges (Pedreira et al., 2003; Ayarza et al., 2004; Fernandez-Viejo et al., 2011; Ruiz et al., 2017), and the effect of Alpine reactivation in the Iberian Peninsula (Teixell et al., 2018; Andrés et al., 2019). In addition to those continental-scale projects, an increased interest in the study of the shallow subsurface has emerged in the last decade in different areas of the Iberian Peninsula. These interests are related to natural resource exploration and exploitation, geo-energy and permanent storage applications, earthquake hazard assessment, or infrastructure planning (Alcalde et al., 2013a, b; Martí et al., 2019).

Here, we report on the readily available database SeisDARE (Seismic DAta REpository), which compiles large geophysical projects developed in the Iberian Peninsula and Morocco. Part of the data comprised in this database were already stored in Geo DB (http://geodb.ictja.csic.es/#dades1, last access: March 2021). However, they lacked a permanent digital object identifier (DOI), thus representing an intermediate step in making the datasets FAIR. SeisDARE goes beyond Geo DB by hosting a larger number of datasets, providing them with a DOI and linking them to scientific literature by means of the Scholix facilities and the DataCite Event Data (Hirsch, 2019, https://search.datacite.org/, last access: March 2021).

The geoscientific data are the basis for the generation of meaningful geological knowledge (Pérez-Díaz et al., 2020), and therefore, the idea behind SeisDARE is to treat the data as any scientific publication, making them as relevant as the publication itself (Carbonell et al., 2020 TS1; DeFelipe et al., 2020). The purpose of this paper is to provide a general overview of the geological setting of the experiments, a brief description of each dataset, and the link to the open-access data. This database is the result of a fruitful national and international inter-institutional collaboration, aiming to enhance multidisciplinary research and to promote advanced research networks. Thus, we are actively working in keeping our database updated, gathering more datasets.

## 2 Outline of SeisDARE

The GEO3BCN database comprises multidisciplinary research data. SeisDARE forms part of this database and is constantly being updated. Especially in the last years, the efforts put into the dissemination of the data have yielded to a significant increase in the number of views and downloads (Fig. 1). The number of view counts has been generally increasing with punctual peaks of views and with an outstanding number of views in August CE4 2020. The number of downloads has strikingly increased since October 2019, reaching the highest value in April 2020, with more than 2500 CE5 downloads. Additional statistics of each item's views and downloads are available and can be consulted directly in the statistics facilities of the repository.

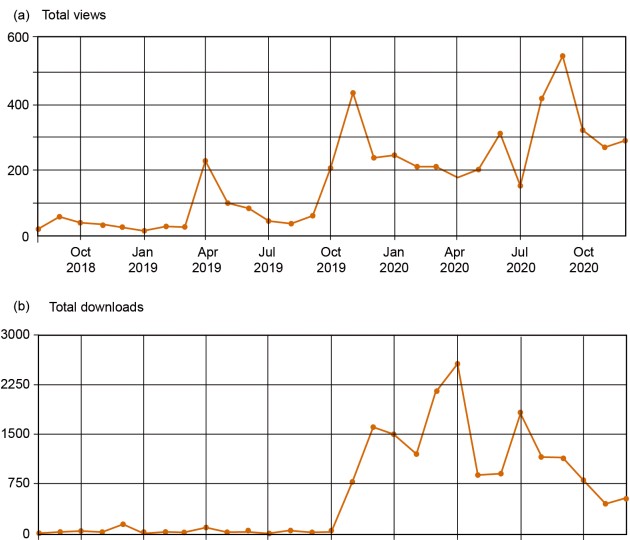

**Figure 1. (a)** Total views and **(b)** total downloads of all datasets from SeisDARE (https://digital.csic.es/handle/10261/101879, last access: January 2021).

SeisDARE contains 21 datasets (last access: January 2021) of DSS (normal incidence, NI, and wide-angle, WA) and HR data (Table 1). Our database comprises the data files together with a comprehensive description of their general characteristics and the acquisition parameters. Furthermore, a list of authors and publications derived from the data are also provided. Additionally, an index card summarizes the information for all projects except for those that are embargoed due to data policy (i.e. usually a reasonable time period for the use of the data by the project members or private companies involved). Those files comprise the general information of the data, a location map of the seismic data, an image of the data as an example, references, and funding agencies. Figure 2 shows an example of the index card of the IBERSEIS-WA project. Ideally, all the content stored in the database is self-explanatory, so that the inexperienced users can handle the datasets straightforwardly.

In general, the philosophy behind SeisDARE is entirely transferable to similar seismic datasets from other areas in the world. The promoters of SeisDARE are actively looking for engagement with other international institutions, either by offering to host their datasets or to serve as a model for the establishment of new similar platforms. This internationalization effort led to the establishment of a collaboration with the Consortium for Continental Reflection Profiling (COCORP), to host the Hardeman County (Texas) dataset. This kind of collaboration adds great value to the database by expanding the range of geological settings sampled and strengthening the research network between different institutions worldwide.

**Table 1.** Seismic datasets comprised in SeisDARE (DSS: deep seismic sounding; HR: high resolution CE6). The location of the seismic datasets is shown in Fig. 3. Note that ESCI, MARCONI, IBERSEIS, ALCUDIA, and VICANAS comprise two or three datasets (see text for details).

| Dataset | Year | Seismic data | Onshore/offshore | Research target | Section | Dataset reference |
|---|---|---|---|---|---|---|
| ILIHA | 1989 | DSS wide angle | Onshore | Iberian Massif | 5.1 | Díaz et al. (2020) |
| ESCI-N ESCI Valencia Trough ESCI Betics | 1991–1993 | DSS normal incidence | Onshore and offshore | Iberian Massif Bay of Biscay Cantabrian Mountains Valencia Trough Betic Cordillera | 5.2 | Pérez-Estaún et al. (2019) Gallart et al. (1993) García-Dueñas et al. (1991) |
| IAM | 1993 | DSS normal incidence | Offshore | Iberian Atlantic Margin | 5.3 | Torné et al. (2018) |
| MARCONI | 2003 | DSS normal incidence and wide angle | Offshore and onshore | Bay of Biscay | 5.4 | Gallart et al. (2019, 2020) |
| IBERSEIS | 2001 and 2003 | DSS normal incidence and wide angle | Onshore | Iberian Massif (South Portuguese Zone, Ossa-Morena Zone and Central Iberian Zone) | 5.5 | Pérez-Estaún et al. (2001a, b) Palomeras et al. (2003) |
| ALCUDIA | 2007 and 2012 | DSS normal incidence and wide angle | Onshore | Iberian Massif (Ossa-Morena Zone and Central Iberian Zone) | 5.6 | Pérez-Estaún et al. (2007) Pérez-Estaún et al. (2012) |
| SIMA | 2010 | DSS wide angle | Onshore | Rif Cordillera and Atlas Mountains | 5.7 | Ayarza et al. (2010a) |
| RIFSIS | 2011 | DSS wide angle | Onshore | Rif Cordillera | 5.8 | Gallart et al. (2011) |
| CIMDEF | 2017 and 2019 | DSS wide angle | Onshore | Duero basin, Central Iberian Zone and Tajo basin | 5.9 | Ayarza and Carbonell (2019) |
| HONTOMÍN | 2010 | HR normal incidence | Onshore | $CO_2$ storage site characterization | 6.1 | Alcalde et al. (2010) |
| VICANAS | 2014 | HR normal incidence | Onshore | Nuclear waste disposal site characterization | 6.2 | Marzán et al. (2013, 2015) |
| INTERGEO | 2015 | HR normal incidence | Onshore | Quaternary seismicity in the Alhama de Murcia Fault | 6.3 | Martí et al. (2015) |
| SOTIEL | 2018 | HR normal incidence | Onshore | Mining exploration in the Iberian Pyrite Belt | 6.4 | Alcalde et al. (2018) |
| COCORP Hardeman County | 1975 | DSS normal incidence | Onshore | Hardeman County (Texas, USA) | 7 | Oliver and Kaufman (1975) |

## 3   Geological setting of the Iberian Peninsula and Morocco

The Iberian Peninsula and Morocco have experienced a long and complex geological history since the Paleozoic, resulting in a rich variety of tectonic and sedimentary domains. These include the Variscan terrains of the Iberian Massif, Pyrenean Axial Zone and Iberian Chain; broad Mesozoic basins; the Alpine mountain belts; and the Cenozoic foreland basins (Fig. 3). In addition, Fig. 4 shows some of the crustal models resulting from the NI and WA datasets belonging to Seis-DARE, covering an almost complete section of the Iberian Peninsula (onshore and offshore) and Morocco.

The Variscan orogen resulted from the convergence of the Laurentia–Baltica and Gondwana continents, yielding to the supercontinent Pangaea (e.g. Matte, 1991, 2001; Murphy and Nance, 1991; Simancas et al., 2003; Pérez-Cáceres et al., 2016). In the Iberian Peninsula, the main Variscan rock outcrop corresponds to the Iberian Massif, and it is divided into

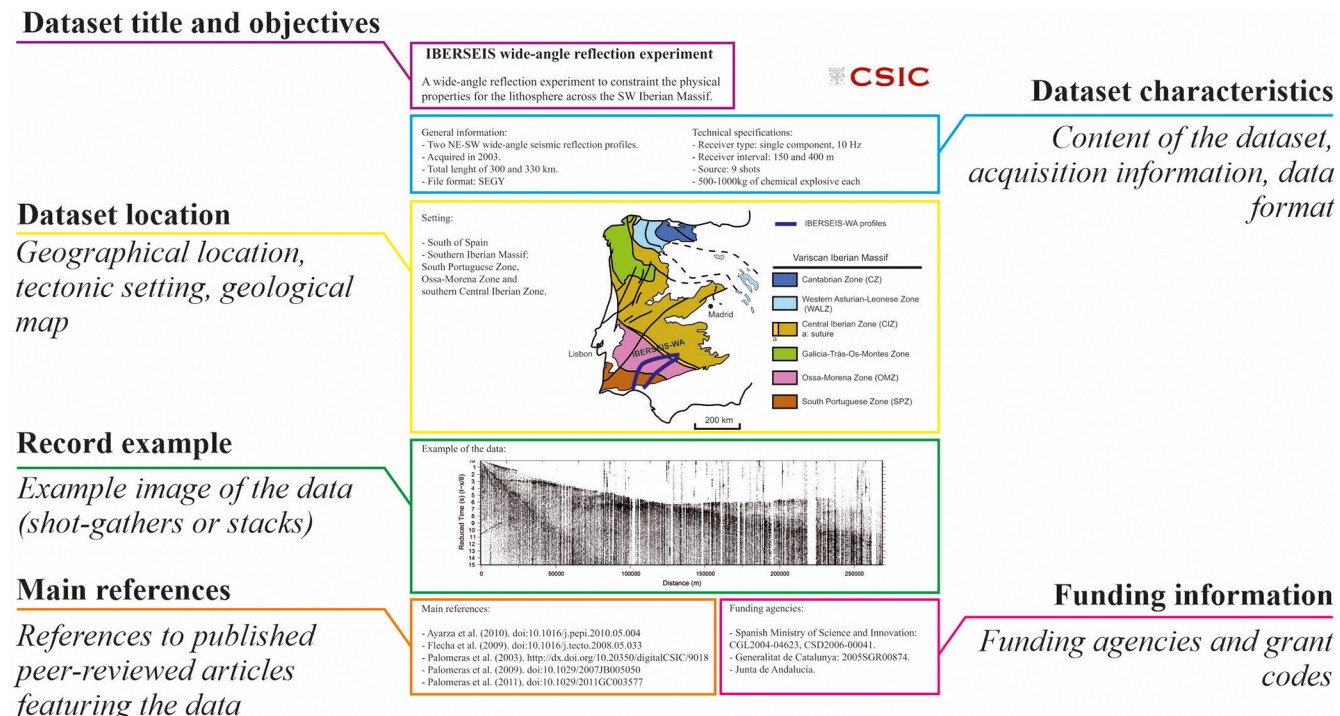

**Figure 2.** Example of the index card containing the IBERSEIS-WA project (Sect. 5.5.2) information.

six zones (Julivert et al., 1972; Fig. 3). The West Asturian-Leonese Zone (WALZ) and the Cantabrian Zone (CZ) are the northernmost zones. The Central Iberian Zone (CIZ) is the largest zone and includes the entirely allochthonous Galicia-Trás-Os-Montes Zone (GTOMZ) towards the northwest (Martínez Catalán et al., 1997; Arenas et al., 2007). South of the CIZ, the Ossa-Morena Zone (OMZ) is a highly deformed area of upper Proterozoic–lower Paleozoic rocks (Matte, 2001; Pérez-Cáceres et al., 2016, 2017). The southernmost zone of the Iberian Massif is the South Portuguese Zone (SPZ) and hosts the largest concentration of volcanic massive sulfide deposits worldwide, the Iberian Pyrite Belt (Tornos, 2006). The CZ and the SPZ represent the external zones of the Variscan orogen whereas the rest represent the internal zones (e.g. Martínez Catalán et al., 1997). The structure of the CZ, WALZ, CIZ, OMZ, and SPZ has been studied through the ESCI-N, IBERSEIS, and ALCUDIA experiments (Fig. 4c and d TS2). In these areas the Alpine inversion resulted mostly in localized faults and well-defined crustal imbrications; thus the data reflect a nearly complete crustal section of the late Variscan orogen. A main feature observed along the Iberian Massif is the poor coupling existing between the upper and lower crustal reflectivity, which has been interpreted as the image of contrasting deformation mechanisms to accommodate shortening at both crustal levels (Simancas et al., 2003, 2013).

Throughout most of the Mesozoic, lithospheric extension in relation to the opening of the Atlantic Ocean and Bay of Biscay gave way to the formation of rift domains of hyperextended crust and exhumed mantle and deep Cretaceous basins in the Pyrenean–Cantabrian realm (e.g. Ziegler, 1988; García-Mondéjar et al., 1996; Jammes et al., 2009; Pedreira et al., 2015; Tugend et al., 2014, 2015; DeFelipe et al., 2017; Ruiz et al., 2017). The mechanisms and geodynamic evolution of the Iberian Atlantic margin and Bay of Biscay were investigated with the pioneer IAM and ESCI-N projects, followed by the MARCONI initiative (e.g. Banda et al., 1995; Álvarez-Marrón et al., 1995a, b, 1996; Pulgar et al., 1995, 1996; Fernández-Viejo et al., 2011; Ruiz et al., 2017). These projects allowed mapping of the offshore distribution of the North Pyrenean basins, to assess the lateral variations in the crust in the North Iberian Margin and to evaluate the inheritance of the extensional structures in the Alpine orogeny (Álvarez-Marrón et al., 1996; Fernández-Viejo et al., 1998; Ferrer et al., 2008; Roca et al., 2011).

From the Late Cretaceous to the Miocene, the Alpine convergence resulted in the tectonic inversion of the Mesozoic basins and mountain building (e.g. Muñoz, 1992; Teixell, 1998; Rosenbaum et al., 2002a; Teixell et al., 2016, 2018; DeFelipe et al., 2018, 2019). Seismic imaging in the Cantabrian Mountains, the North Iberian Margin and Duero basin area allowed the interpretation of a crustal thickening under the Cantabrian Mountains, with the Iberian crust subducting towards the north (Pulgar et al., 1996; Gallastegui, 2000; Gallart et al., 1997a; Pedreira et al., 2003, 2015; Gallastegui et al., 2016). More recently, Teixell et al. (2018) sug-

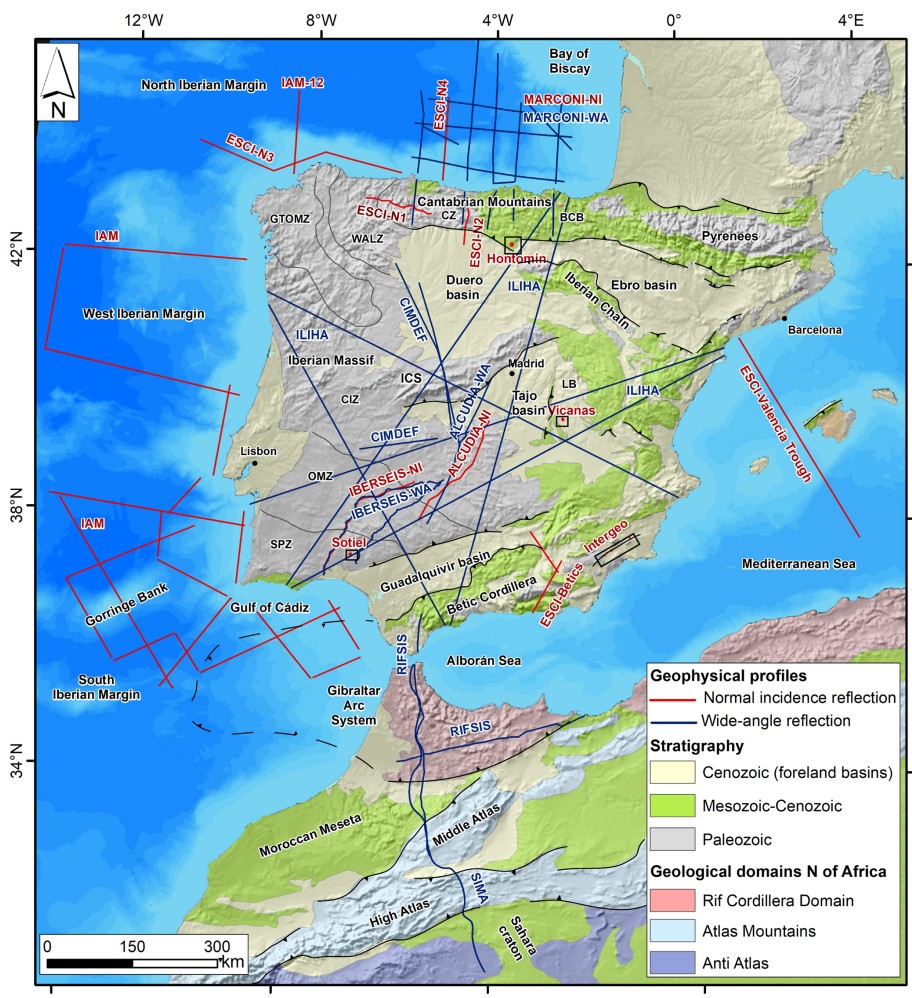

**Figure 3.** Geological map of the Iberian Peninsula and north of Africa with the seismic profiles provided in SeisDARE. The geological units are simplified from the IGME Geological Map of the Iberian Peninsula 1 : 1 000 000 (Rodríguez Fernández et al., 2014) and the bathymetry and topography from OpenTopography (Ryan et al., 2009) and Natural Earth. Note that the MARCONI offshore experiment acquired simultaneously normal incidence and wide-angle reflection, whereas on land, only wide-angle data were acquired. BCB: Basque-Cantabrian basin; CZ: Cantabrian Zone; WALZ: West Asturian Leonese Zone; GTOMZ: Galicia-Trás-Os-Montes Zone; CIZ: Central Iberian Zone; OMZ: Ossa Morena Zone; SPZ: South Portuguese Zone; ICS: Iberian Central System; LR: Loranca Basin.

gested the southwards subduction of the Bay of Biscay crust along the MARCONI-1 profile, although of probable limited extent at this longitude. Towards the west, along the IAM-12 profile, the oceanic crust of the Bay of Biscay subducts towards the south (Teixell et al., 2018). This subduction is also imaged in the ESCI-N3.2 and ESCI-N3.3 profiles as 3D inclined subcrustal reflections (Ayarza et al., 2004).

Towards the south of the Iberian Peninsula, the Gibraltar Arc System comprises the Betic Cordillera and the Rif, the westernmost belt of the Alpine chains (Fig. 3). It is a broad arcuate collision zone separated by the Alborán Sea that resulted from the convergence between the Eurasian and African plates since the Miocene. This area represents a complex tectonic scenario with periods of compression overprinted by extensional episodes and dextral strike-slip movements of the Iberian subplate (e.g. Sanz de Galdeano, 1990; Carbonell et al., 1997; Rosenbaum et al., 2002b; Platt et al., 2013). South of the Rif, the Atlas Mountains were also formed as a consequence of the Cenozoic-to-present convergence between Eurasia and Africa. The Moroccan part of this orogen is formed by two branches, the High and Middle Atlas, that correspond to inverted Mesozoic basins (e.g. Arboleya et al., 2004). The RIFSIS and SIMA projects investigated these areas, sampling the African lithosphere down to the Moroccan Atlas. The topography of the latter is too high when considering the context of limited orogenic shortening featured in this area. Volcanism and intermediate depth seismicity support a model where isostasy is not enough to maintain the orogenic load; thus dynamic topography must have played a key role (Ayarza et al., 2005; Teixell et al.,

2005, 2007). The RIFSIS and SIMA experiments shed some light on the P-wave velocity structure of the lithosphere and the Moho topography in this large area, improving existing models (Ayarza et al., 2014; Gil et al., 2014; Fig. 4e TS3).

## 4 Technical aspects of the data

The datasets available in SeisDARE comprise controlled-source DSS and HR data. All these datasets feature specific acquisition parameters which are detailed in their corresponding bibliography. This section provides a brief summary of the main acquisition characteristics of the different datasets.

For instance, four types of sources were used in the acquisition of the on-land data. For DSS data (i) most of the WA experiments used controlled explosions with charges ranging from 500 to 1000 kg; (ii) large quarry blasts were used as energy sources in the acquisition of the ILIHA project; (iii) for most of the NI experiments vibroseis trucks (4, 22, or 32 t) were used, although ESCI-N1 and ESCI-N2 were acquired using explosions of 20 kg. For HR data, two types of sources were used: (i) 250 kg accelerated-weight drop and (ii) 15–32 t vibroseis trucks. Regarding the DSS offshore data, the energy sources were airgun arrays with 2000 psi (13.85 Mpa) with different ranges of volumes depending on the research vessels used in each experiment: MV *Seisquest* with a volume of 5490 in.$^3$ CE7 (in ESCI-N), the MV *Geco Sigma* with a volume of 7542 in.$^3$ CE8 (in IAM), and the BIO *Hespérides* with volumes ranging from 2690 to 1435 in.$^3$ CE9 (in MARCONI).

The receivers onshore consisted of geophones of one or three components with sensors from 10 Hz to 20 s. A number of wide-angle experiments (IBERSEIS-WA, ALCUDIA-WA, SIMA, RIFSIS, and CIMDEF) used a relatively large number of receivers, equipped with RefTek 125A instruments (also known as TEXANS) that belong to the IRIS-PASSCAL instrument pool. In addition, the CIMDEF experiment also used SERCEL-Unite data loggers CE10 equipped with 10 Hz geophones and WorldSensing SPIDER data loggers with 2 Hz sensors. For the MARCONI-WA experiment, RefTek 72A and Leas-Hathor data loggers with sensors of 1, 5, and 20 s were used. The offshore receivers used in the NI experiments were hydrophone streamers with different lengths. Finally, the WA experiments used ocean-bottom seismometers or hydrophones (OBSs or OBHs) as recording devices.

The data included in SeisDARE are in SEG-Y, SEG-D (https://seg.org/Publications/SEG-Technical-Standards, last access: March 2021; Barry et al., 1975), or XDR SU format (https://wiki.seismic-unix.org/doku.php, last access: March 2021; Forel et al., 2008), the most commonly used formats for seismic data. The older datasets (ILIHA, ESCI, and IAM) were recorded on magnetic track tapes and were successfully recovered and transformed to readable digital formats

(SEG-Y or SU). The datasets are accessible as raw data, and processed stack cross sections (migrated or not) are also available for some datasets. Raw data will fulfil the needs of researchers who would like to develop new processing approaches, and the already processed data will provide images for scientists aiming to test alternative interpretations.

## 5 DSS datasets

This section describes the DSS datasets hosted in SeisDARE, ordered by year of acquisition, that sampled different parts of the Iberian Peninsula and Morocco. Each subsection provides a description of the individual datasets and the most relevant references associated.

### 5.1 ILIHA

The pioneer ILIHA (Iberian LIthosphere Heterogeneity and Anisotropy) project was acquired in 1989 and consisted of a star-shaped arrangement of six long-range wide-angle reflection profiles covering the entire Iberian Peninsula. It was designed to study the lateral heterogeneity and anisotropy of the deep lithosphere of the Variscan basement in the Iberian Peninsula. The results of this project suggested a layered lower lithosphere reaching at least 90 km depth, contrasting with the heterogeneity of the Variscan surface geology (Díaz et al., 1993, 1996).

### 5.2 ESCI

The ESCI project (Estudios Sísmicos de la Corteza Ibérica – Seismic Studies of the Iberian Crust) was conducted to obtain multichannel deep seismic reflection images in three areas of the Iberian Peninsula (Fig. 3): the NW of the Iberian Peninsula (ESCI-N), the Gulf of Valencia (ESCI-Valencia Trough), and the Betic Cordillera (ESCI-Betics).

#### 5.2.1 ESCI-N

ESCI-N comprises four multichannel seismic experiments, some of them also including wide-angle reflection datasets. ESCI-N1 is an onshore profile that runs E–W through the Variscan structure of the WALZ and the CZ. ESCI-N2, also onshore, runs with a N–S orientation and was aimed to image the Alpine imprint of the Cantabrian Mountains. ESCI-N3 is made up of three seismic profiles that follow a swerved line in a mainly E–W direction in Galicia and west Asturian offshore. Finally, ESCI-N4 runs in a N–S direction offshore (Álvarez-Marrón et al., 1995a, b, 1996, 1997; Gutiérrez-Alonso, 1997).

Beneath the CZ, ESCI-N1 shows the presence of a basal gently west-dipping detachment identified as the Cantabrian Zone basal detachment. Towards the WALZ, a stack of thrust sheets probably affecting Precambrian rocks is placed above a crustal-scale ramp dipping to the west (Pérez-Estaún et al.,

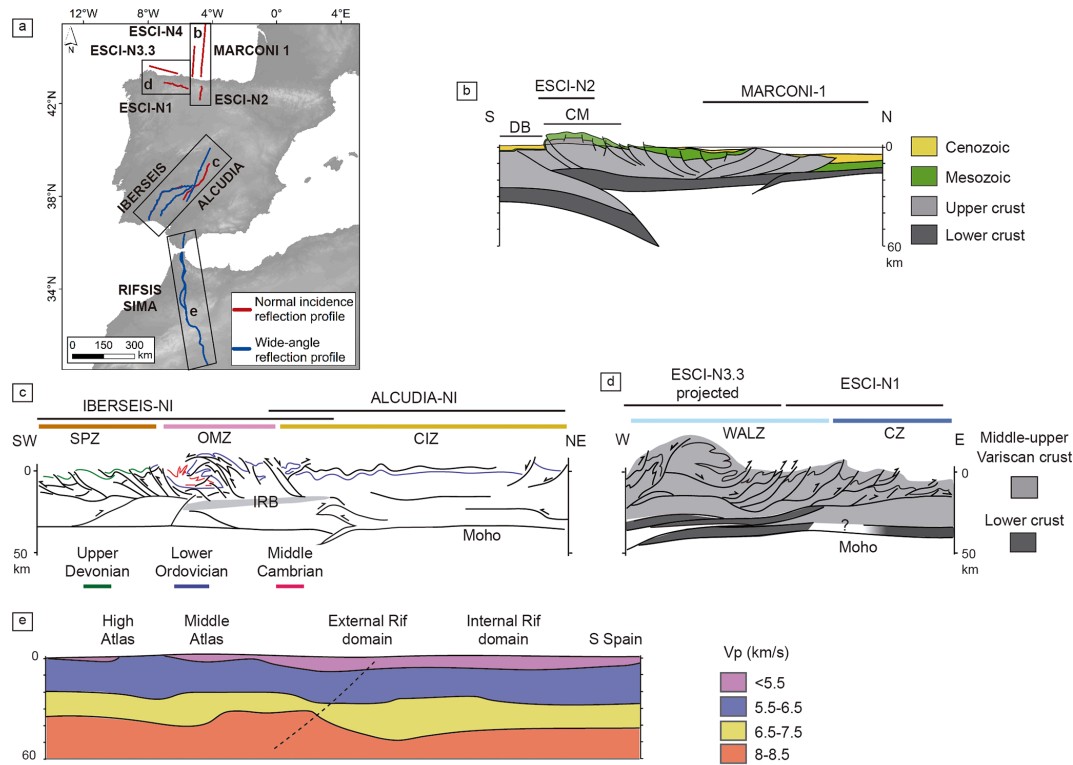

**Figure 4. (a)** Map of the Iberian Peninsula and north of Africa with the location of the ESCI-N, MARCONI 1, IBERSEIS, ALCUDIA, RIFSIS, and SIMA projects. The topography is from OpenTopography (Ryan et al., 2009); **(b)** crustal structure of the Cantabrian Mountains (CM), Duero basin (DB), and Bay of Biscay based on ESCI-N and MARCONI data (after Gallastegui, 2000; Pedreira et al., 2015; Gallastegui et al., 2016; Teixell et al., 2018); **(c)** crustal structure of the southern Iberian Massif (SPZ: South Portuguese Zone; OMZ: Ossa Morena Zone; and CIZ: Central Iberian Zone; IRB: Iberian Reflective Body) revealed by the IBERSEIS and ALCUDIA projects (after Simancas et al., 2013); **(d)** crustal structure of the northern Iberian Massif revealed by the ESCI-N1 and N3.3 profiles (after Ayarza et al., 1998; Fernández-Viejo and Gallastegui, 2005; Simancas et al., 2013); and **(e)** P-wave velocity model obtained from the wide-angle reflection data of RIFSIS and SIMA (simplified from Ayarza et al., 2014; Gil et al., 2014). TS4

1994, 1997; Gallastegui et al., 1997; Fernández-Viejo and Gallastegui, 2005). The ESCI-N3 depicted a Moho offset between the CIZ and the WALZ, and the southward subduction of the oceanic crust of the Bay of Biscay as a consequence of Alpine shortening (Martínez Catalán et al., 1995; Álvarez-Marrón et al., 1995a, b, 1996; Ayarza et al., 1998, 2004). Furthermore, ESCI-N2 and ESCI-N4 revealed a crustal thickening beneath the highest summits of the Cantabrian Mountains which was interpreted as the northward subduction of the Iberian crust under this mountain range (Pulgar et al., 1995, 1996; Fernández-Viejo and Gallastegui, 2005).

### 5.2.2 ESCI Valencia Trough

The Valencia Trough is a NE–SW-trending Cenozoic basin in the western Mediterranean Sea bounded by the Betic Cordillera to the southwest, the Iberian Chain to the west and the Pyrenees to the north (Fig. 3). Different geodynamic models were proposed for the Valencia Trough during the 1980s and early 1990s, but there was a lack of agreement on its crustal structure and geodynamic evolution. Therefore,

ESCI Valencia Trough was conducted to shed light on the structure of the western Mediterranean, providing a continuous image of this area for the first time. This experiment revealed that beneath the Valencia Trough the crust has continental affinity and is strongly attenuated. The Moho is located at 17–18 km depth, thus being the strongest thinning reported so far in comparable passive margins (Gallart et al., 1994, 1997b).

### 5.2.3 ESCI Betics

The ESCI Betics dataset consists of two profiles in the southeast of Spain through the Betic Cordillera (Fig. 3). The ESCI Betics 1 is a 90 km long NW–SE profile that crosses the Guadalquivir basin, the external part of the Betic Cordillera, and the Neogene–Quaternary basin at the limit with the internal zone. The ESCI Betics 2 profile runs for 160 km across the emerged part of the Alborán domain with a NNE–SSW orientation (Carbonell et al., 1997; Vegas et al., 1997). The main aim of this project was to image the structure of the crust, as well as to investigate the development of collisional

structures, the response to extensional stresses in regions of recently thinned crust, and the correlation between the crustal structure and the distribution of seismicity in a tectonically active area (García-Dueñas et al., 1994). It provided the first complete structural transect of the northern flank of the Gibraltar Arc and the Alpine metamorphic complexes of the Betic Cordillera.

## 5.3   IAM

The IAM (Iberian Atlantic Margin) project comprised 19 offshore reflection seismic profiles covering a total length of ∼ 3800 km. This project was set to study the deep continental and oceanic crusts in different parts of the Iberian Atlantic margin, thus sampling the north, west, and south Iberian margins (Banda et al., 1995; Torné et al., 1995, 2018; Fig. 3).

The North Iberian Margin was imaged by a N–S profile in the western part of the Bay of Biscay (Fernández-Viejo et al., 1998). It revealed the crustal structure in this area and its nature for the first time. Results indicate that the sedimentary infill of the abyssal plain is probably underlain by an oceanic basement (Álvarez-Marrón et al., 1997). The Moho is imaged as a subhorizontal reflection package located on land at 23–26 km (Córdoba et al., 1987) that features a rapid shallowing up to 15 km depth in the abyssal plain.

The West Iberian Margin is structurally divided in four zones: oceanic crust, peridotite ridge, transition zone, and thinned continental crust. The crust decreases its thickness progressively to the west towards an oceanic crust of 6.5–7 km thick (Pickup et al., 1996; Sutra and Manatschal, 2012; Sutra et al., 2013). The peridotite ridge is a basement high that coincides with the eastward boundary of the seafloor-spreading magnetic anomalies (Dean et al., 2000). The transition zone has been interpreted as a tectonically exposed upper mantle, extensively serpentinized, that continues laterally to the continental crust.

The Southern Iberian Margin has undergone a complex tectonic history through a sequence of Mesozoic rifting and collisional events associated with the Africa–Eurasia plate convergence (González et al., 1996). This area corresponds to the Azores–Gibraltar seismic zone and comprises two regions: the Gorringe Bank and the Gulf of Cádiz. The Gorringe Bank is characterized by an irregular topography and a large-amplitude gravity anomaly (Souriau, 1984; Gràcia et al., 2003; Cunha et al., 2010). It consists of mafic and ultramafic plutonic and extrusive rocks covered by Aptian to Pliocene sediments (Tortella et al., 1997; Jiménez-Munt et al., 2010). On the other hand, the basement of the Gulf of Cádiz is a continental crust that thins progressively from east to west (González-Fernández et al., 2001). The transition from the continental crust of the Gulf of Cádiz to the transitional/oceanic crust of the Gorringe Bank is imaged as a series of E–W-oriented thrusts, folds, and diapirs formed during the Cenozoic convergence (Tortella et al., 1997).

## 5.4   MARCONI

The deep seismic survey MARCONI (MARgen COntinental Nord Iberico – North-Iberian Continental Margin) was carried out to understand the processes that governed the evolution of the Bay of Biscay and to establish the lithospheric structure of its southeastern part (Gallart et al., 2004). It included the simultaneous acquisition of 11 multichannel and wide-angle deep seismic reflection profiles with a total length of 2000 km (Fernández-Viejo et al., 2011; Fig. 3). This project imaged the sedimentary architecture in this part of the Bay of Biscay where no boreholes or direct geological data were available. The results of the MARCONI project confirmed the existence of compressive N-directed structures related to the Alpine orogeny and developed at the foot of the continental slope, as well as the existence of NNE–SSW-oriented transfer zones. Furthermore, it helped to identify the Cretaceous rift domains. The analysis of the velocity–depth profiles and the multichannel seismic data allowed the conclusion that the SE part of the Bay of Biscay is formed by a stretched/thinned continental crust with the Moho located between 30 and 20 km, the NW part of the study area is interpreted as a transitional crust from continental to oceanic with the Moho identified at less than 20 km, and the on-land part of the experiment (imaged only with the wide-angle experiment) showed the continental crust thickening as a consequence of the Alpine orogeny (Ferrer et al., 2008; Fernández-Viejo et al., 2011; Roca et al., 2011; Ruiz, 2007; Ruiz et al., 2017).

## 5.5   IBERSEIS

In SW Spain, a NI profile and two WA seismic profiles were acquired in 2001 and 2003 respectively to image the southernmost part of the Iberian Massif in the framework of the IBERSEIS project (Fig. 3).

### 5.5.1   IBERSEIS-NI

IBERSEIS-NI is a 303 km long NE–SW-oriented seismic reflection profile that sampled three major tectonic domains: the SPZ, the OMZ, and the southernmost part of the CIZ (Figs. 3 and 4c; Simancas et al., 2003, 2006). The high-resolution image obtained matched existing geological cross sections and indicated the presence of left-lateral strike-slip displacements of Carboniferous age in the boundaries of the OMZ. The profile also provides an image of the alleged suture between the OMZ and the SPZ, which is formed by a complex accretionary prism of sediments and oceanic basalts. The limit between the OMZ and the CIZ is assembled as an accretionary wedge of high-pressure metamorphic rocks. Fold-and-thrust belts in the upper crust merge downwards in a decoupling horizon between the upper and the lower crust. In the OMZ and CIZ, a 175 km long thick band of relatively high amplitude, the Iberian Reflective Body

(IRB), is interpreted as a layered mafic–ultramafic body intruded along a mid-crustal decollement. In the entire transect, a horizontal Moho reflection at 10.5 s suggests a 30–33 km thick crust (Simancas et al., 2003; Carbonell et al., 2004; Simancas et al., 2006).

### 5.5.2 IBERSEIS-WA

The IBERSEIS-WA profile was acquired with the aim of providing constraints on the physical properties of the lithosphere beyond the results obtained in the IBERSEIS-NI profile. The IBERSEIS-WA project consisted of two transects running subparallel in a NE–SW to ENE–WSW direction. Transect A coincides with the trace of the IBERSEIS-NI profile, and transect B was located farther to the east. Both transects join at their northern end. The main results of this project provided the P- and S-wave velocity models of the crust and upper mantle in this region for the first time. These experiments also allowed modelling of the density (Palomeras et al., 2011a) and the Poisson's ratio (Palomeras et al., 2011b), which helped to provide an additional petrological description to estimate the crustal composition. Furthermore, it allowed the location of the crust–mantle boundary at a depth of 31–33 km and identification of a subcrustal discontinuity at 66–70 km (Palomeras et al., 2009), later interpreted as the Hales discontinuity or gradient zone (Ayarza et al., 2010a).

### 5.6 ALCUDIA

The ALCUDIA dataset consists of two experiments, NI and WA, acquired in 2007 and 2012, respectively. These two experiments sampled the CIZ from the boundary with the OMZ to the northern part of the Tajo basin in approximately a NE–SW direction. It constitutes the northward prolongation of the previously acquired IBERSEIS seismic profiles (Figs. 3 and 4c).

### 5.6.1 ALCUDIA-NI

ALCUDIA-NI is a 230 km long NE–SW profile that imaged the lithospheric architecture of the Variscan orogen in the CIZ (Martínez Poyatos et al., 2012). The processed ALCUDIA-NI seismic profile shows a Moho discontinuity at ∼ 10 s TWT (two-way time, ca. 30 km depth) that overlies a mantle where subhorizontal reflectivity was identified between 14 and 19 s TWT. The most prominent feature is a lower crust tectonic wedge in the southern segment of the transect, causing a local crustal thickening. The interpreted structures, as deduced from surface geology and the seismic image, show that deformation was distributed homogeneously in the upper crust, whereas it was concentrated in wedge and thrust structures at specific sectors in the lower crust (Martínez Poyatos et al., 2012; Simancas et al., 2013).

### 5.6.2 ALCUDIA-WA

The ALCUDIA-WA experiment aimed to constrain the lithospheric structure and resolve the physical properties of the crust and upper mantle in the CIZ. This profile has led to a well-constrained 280 km long and 50 km deep P-wave velocity model. Its major contribution was to confirm the existence of an incipient crustal thickening towards the Tajo basin. High velocities towards the northern part of the profile correspond to igneous and/or high-grade metamorphic rocks in agreement with the geology of the Iberian Massif, in the area close to the south of the Iberian Central System (Ehsan et al., 2015). In addition, a new interpretation of these data provides a S-wave model and Poisson's ratio values which allowed for a more accurate model of the nature of the crust (Palomeras et al., 2021).

### 5.7 SIMA

SIMA (Seismic Imaging of the Moroccan Atlas) is a wide-angle reflection seismic experiment that runs through the Rif, Middle and High Atlas, and Sahara craton of Morocco. This project provided the P-wave velocity structure of the crust and the geometry of the Moho boundary. Final models image the Atlas limited crustal root, which is defined by the northward imbrication of the Sahara crust underneath the Atlas itself. The limited extension of the crustal root supports the need of dynamic topography models to explain the current Atlas topography (Ayarza et al., 2014). In addition, the low P-wave velocities obtained in the lower crust and mantle of the Middle and High Atlas were assigned to the existence of high temperatures and partial melts at these levels, probably as a consequence of the Cenozoic-to-present magmatic activity of the Atlas region due to mantle upwelling. These low P-wave velocities might indicate slightly lower densities that could modify the existing gravity models (Ayarza et al., 2005).

### 5.8 RIFSIS

The RIFSIS wide-angle reflection seismic experiment consists of two E–W- and N–S-oriented profiles, 330 and 430 km long respectively. This acquisition geometry was designed to target the significantly low Bouguer anomaly associated with this mountain range (Hildengrand et al., 1988). This project, which also aimed to improve the geodynamic models of the Gibraltar Arc, identified a 50 km deep crustal root below the Rif that thins rapidly towards the east to 20 km in the contact between the external part of the Rif with the former North African margin (Gil et al., 2014). Jointly, RIFSIS and SIMA provided a 700 km long profile from the southernmost Iberian Peninsula to the northern Sahara desert (Carbonell et al., 2013, 2014 TS5).

## 5.9 CIMDEF

The most recent wide-angle reflection–refraction experiment in the Iberian Peninsula was acquired as part of the CIMDEF project (Andrés et al., 2019; Carbonell et al., 2020 TS6). This project aimed to obtain a P-wave velocity model of the western part of the Duero basin, the CIZ, including that of the Iberian Central System and the Tajo basin. This profile was meant to fill the data gap between the ALCUDIA and IBER-SEIS profiles in the south and the ESCI-N profiles in the north (Fig. 3). The dataset was acquired along two time periods: the seismic survey carried out in 2017 consisted of two profiles, NNW–SSE- and E–W-oriented of ca. 275 and 135 km length respectively. The survey carried out in 2019 consists of a NW–SE 340 km long profile. This dataset is embargoed until 1 January 2024.

## 6 HR seismic dataset

In the 2010s, HR reflection seismic experiments were focused on shallow subsurface targets. These experiments were acquired with different objectives, including reservoir characterization for carbon capture and storage (CCS), nuclear waste storage, mineral resource exploration, and seismic hazard assessments. These experiments involved imaging with both controlled and natural seismic sources and provided images and velocity models of the subsurface, from approximately 50 to 2000 m depth.

### 6.1 HONTOMÍN

To characterize the first Spanish research facility for geological storage of $CO_2$, a 3D reflection seismic experiment was carried out in Hontomín (Fig. 3). This area is located in the southwestern part of the Basque–Cantabrian basin (Fig. 3), which developed as a thick Mesozoic basin, tectonically inverted during the Alpine orogeny (e.g. DeFelipe et al., 2018, 2019). This experiment targeted a saline aquifer located at 1450 m depth within Lower Jurassic carbonates with a main seal formed by interlayered Lower to Middle Jurassic marl and limestone. The 3D geological structure consists of an asymmetric dome crosscut by a relatively complex fault system (Alcalde et al., 2013a, b, 2014). This study set the basis for the first CSS pilot plant in Spain. The dataset is embargoed until 1 April 2022 CE11.

### 6.2 VICANAS

The increased need of facilities for temporal and long-term storage of radioactive waste has encouraged new geological and geophysical projects to characterize the structure of suitable settings. Within the framework of the VICANAS project, four reflection seismic profiles and a 3D HR seismic tomography dataset were acquired in the Loranca Basin (Fig. 3). The 2D seismic reflection profiles were intended to characterize the shallow subsurface (up to 1000 m) at the regional level, focusing on faults and fracture networks that could potentially affect the stability of the waste disposal site. The HR seismic tomography survey provided a full 3D P-wave seismic velocity image of an area of 500 m × 500 m (Marzán et al., 2016, 2019). This experiment was specially designed to image the upper 100 m that directly interacts with the ongoing construction works. The seismic tomography results combined with geophysical measurements from boreholes and 2D electrical resistivity tomography profiles provided a detailed mapping of the different lithology contacts that build up the sedimentary sequence filled up to 200 m of fluvial and lacustrine facies sediments (Álvarez-Marrón et al., 2014; Martí et al., 2019; Marzán et al., 2019). Within the scope of this project, Marzán et al. (2021) reinterpreted the 3D HR seismic dataset to provide a more consistent 3D P-wave velocity model by integrating the resistivity model with well-log data. To jointly interpret these data, the authors developed a machine-learning scheme that resulted in a 3D lithological model highly well correlated with the known geology.

### 6.3 INTERGEO

The INTERGEO dataset aims to characterize the seismogenic behaviour of active faults in strike-slip tectonic contexts. The case study of this project is the Alhama de Murcia Fault (AMF), located in the Betic Cordillera (Fig. 3). This fault was responsible for one of the most destructive recent earthquakes in Spain (9 fatalities, 300 injuries, and serious damages), occurring in May 2011 in Lorca, with a moment magnitude, $M_w$ TS7, of 5.2 (Martínez-Díaz et al., 2012). This earthquake was triggered by a rupture area of 3 km × 4 km along the AMF in a transpressive context. Thus, in order to image the complex structure at depth of the southwestern part of the AMF, six NW–SE-oriented reflection seismic profiles were successfully acquired. The seismic reflection profiles and travel time tomography allowed the identification and characterization of the contact between the Miocene–Pliocene detrital sediments and the basement, the internal structure of the AMF, and its different branches (Gascón Padrón, 2016; Ardanaz et al., 2018).

### 6.4 SOTIEL

The European Institute of Technology through the RawMaterials programme has encouraged the development of cost-effective, sustainable, and safe research and innovation solutions for mineral exploration. Within this framework, the SIT4ME project has implemented seismic mineral exploration methods at a reduced cost, analysing the efficiency and capabilities of controlled-source (e.g. reflection imaging techniques) and natural-source (e.g. ambient noise interferometry) methods in mining areas. For this purpose, a multi-method seismic dataset was acquired in the Sotiel-Coronada

mine in the Iberian Pyrite Belt (SPZ) to image a massive sulfide ore body intruded in volcanic and siliciclastic rocks at a depth of 300–500 m (Alcalde et al., 2019). This dataset is embargoed until 1 April 2022.

## 7   Enlarging borders in DSS data sharing: COCORP Hardeman County, Texas

The Consortium of Continental Reflection Profiling (COCORP, http://www.geo.cornell.edu/geology/cocorp/COCORP.html, last access: March 2021) was a research group that pioneered seismic reflection profiling of the crust and upper mantle. COCORP worked during the 1970s and 1980s in the acquisition of more than 8000 km of seismic profiling in the USA (Brown et al., 1987) and stimulated major deep seismic exploration programmes in over 20 countries, such as ECORS in France, DEKORP in Germany or LITHOPROBE in Canada. COCORP demonstrated how seismic reflection information on the geological basement can contribute to address questions in resource exploration (Brown, 1990). In its first experiment COCORP acquired 37 km of common-midpoint (CMP) stacked seismic reflection profiles in Hardeman County, Texas (Oliver et al., 1976; Schilt et al., 1981). These data imaged the Cambrian to Permian sedimentary rocks lying unconformably over a Precambrian basement intruded by relatively homogeneous igneous plutons (Oliver et al., 1976). Prominent layering in the crystalline basement was first imaged in Hardeman County and was later found to underlie much of the east central USA, perhaps representing major igneous intrusions on a continental scale (Kim and Brown, 2019). Furthermore, COCORP Hardeman County proved the viability of using the vibroseis technique to characterize the geological structure of the deep crust, resulting in a widespread technology of acquisition today (Finlayson, 1975).

## 8   Data availability

The SeisDARE is available in the Spanish National Research Council repository, DIGITAL.CSIC (https://digital.csic.es/handle/10261/101879 TS8, last access: March 2021). A detailed list of the datasets can be found in Table 1.

## 9   Conclusions and final considerations

Seismic reflection data (normal incidence and wide angle) provide a high-resolution image of the lithosphere and set constraints on the structure and nature of the deep and shallow subsurface. The controlled-source seismic data are useful not only in basic science but also in applied science, like in resource exploration and natural hazard assessment. Deep seismic sounding data (DSS) data are expensive and logistically complex to acquire and often require a huge scientific effort involving several national and international research groups. The reproducibility of scientific results is dependent on the availability of data, which reinforces the paradigm of transparent and open-access data-driven science as well as fosters innovation. Legacy data can be useful by themselves or by applying the latest innovative processing techniques to generate new meaningful outputs. SeisDARE has been developed to facilitate the preservation and reuse of the existing data by future generations of geoscientists by hosting seismic data in the online institutional repository DIGITAL.CSIC. The SeisDARE database accomplishes the international mandates of open-access data and the FAIR principles of data management. It is the result of a close collaboration between national and international institutions and encourages new networks to make seismic data easily available. Currently, SeisDARE contains 21 seismic datasets of deep seismic sounding and high-resolution data acquired since the 1980s in the Iberian Peninsula and Morocco. In addition, as a result of this internationalization effort we established a collaboration with the scientists that led the Consortium for Continental Reflection Profiling (COCORP) to host the pioneering Hardeman County dataset. All these datasets aimed to characterize different geological settings, ranging from the continental-scale Variscan and Alpine orogens and the offshore Iberian Margins to exploration-scale studies. SeisDARE is being constantly updated and is accessible via a web browser. It is free and open, bringing endless research and teaching opportunities to the scientific, industrial, and educational communities.

**Author contributions.**   IDF, JA, MI, DM, MR, IM, JD, PA, IP, JLFT, RR, and RC worked on the data acquisition, compilation, collation, and dissemination of the datasets. CM and IB provided the facilities to upload the datasets into DIGITAL.CSIC. LB provided additional data to enlarge our database. IDF prepared the manuscript with contributions from all co-authors.

**Competing interests.**   The authors declare that they have no conflict of interest.

**Acknowledgements.**   We acknowledge support of the publication fee by the CSIC Open Access Publication Support Initiative through its Unit of Information Resources for Research (URICI). We thank Florian Haslinger and the anonymous reviewer for their constructive comments that greatly improved the manuscript.

**Financial support.**   This research has been supported by the European Plate Observation System (grant nos. EPOS IP 676564 and EPOS SP 871121), EIT Raw Materials (grant no. 17024 SIT4ME), the Seismology and Earthquake Engineering Research Infrastructure Alliance for Europe (grant no. SERA 730900), and the Ministerio de Ciencia e Innovación (Juan de la Cierva fellowship – IJC2018-036074-I).

**Review statement.** This paper was edited by Kirsten Elger and reviewed by Florian Haslinger and one anonymous referee.

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

## Remarks from the language copy-editor

## Remarks from the typesetter