# Peer review of "Reassessing the lithosphere: SeisDARE, an open access seismic data repository"

_Earth System Science Data, 2020_

## Referee Comment (RC1) · Anonymous Referee #1 · 25 Oct 2020

GENERAL COMMENTS

In manuscript essd-2020-208 the authors present a database (SeisDARE) with data of controlled source seismic experiments. Currently, the database contains 19 datasets, most of them covering the Iberian Peninsula and Morocco; it is open for more datsets. The data itself are stored in and accessible at CSIC's GEO3BCN database. Each dataset consists of the metadata and the data in standard format (file-based, i.e. SEG-Y). A DOI is attached to each dataset. The metadata of each data set (including authors/creators, title, characteristics, data examples, references and funding information) is visualized as "landing pages" in the web-browser and so-called index cards (human readable); it can also be accessed in different (machine-readable) formats.

I congratulate the authors for putting together this nice collection of seismic datasets,

to preserve them and to make them public for further use following the FAIR idea. The datasets provide a unique opportunity to get insights into the deeper structure of the Iberian Peninsula/Morocco and the processes involved in forming them. Crustal-scale seismic profiles are usually very expensive and cannot easily be reproduced. So, keeping and distributing the datasets in this structured way is of high value and is highly interesting for future research (and potentially other purposes).

The article is a good way to support the publication of the datasets. The article is logically structured and reads very easy. References for the individual datasets seem complete (see also one of the issues below). I have found only very few typos (see below). The figures are informative and of high quality as is the whole manuscript. A discussion of the accuracy, calibration, processing, etc. are not in detail given in the article, but references have been made to the original articles (in which these aspects should be presented).

So I think the topic is extremely well suited to be presented in ESSD I found some issues which I feel the authors should address before the manuscript is eventually ready for publication (moderate revision).

SPECIFIC COMMENTS:

- The technical details describing the archived data should be somewhat extended (it seems that the majority of the metadata is to large extend related to the bibliographical data such as authors, title etc.). Actually this is an issue for both the repository and the article. Most of the data (I have checked a few) are stored in SEG-Y format, which is a standard exchange format, however, I suggest to add (in addition to the existing reference/link to SEG) a reference to e.g. Barry, K.M.; Cavers, D.A.; Kneale, C.W. (1975): Recommended standards for digital tape formats. Geophysics 40 (2): 344–352). Other datasets are stored in seismic unix (su) format and I think a link or at least a short description or statement regarding this de facto standard format should be added to the article (e.g., https://wiki.seismic-unix.org/doku.php, but there might be

better references). In line 120 it says that "Data are mainly in SEG-Y format...". Since datasets in su format is also found, could you be more complete and specific? Is there a policy of the GEO3BCN repository which formats to use? Can you state more specifically which quality control is performed at the GEO3BCN repository? Any checks or guidelines which header words are set or should be provided? Especially for su-files it would be good to know whether it is "old su" format or XDR format, and whether it is little or big endian. I suggest a subsection "Data format" or "Technical issues of the data" in the article which could have at least some general statement regarding these issues (or where I can find this information in the website of the GEO3BCN repository).

- Although I think that su-format is quite handy and widely accepted, it is actually not a standardized exchange format... Actually SEG-Y should be preferred...

- [more a general comment] I did some spot checks of this large collection of data sets (only spot checks because the data is many hundreds of Gbyte (or even Tbyte); some of the datsets are even embargoed and thus not available). Data conversion and displaying was quite easy e.g. for the IBSERSEIS NI and ILIHA data. Nevertheless, it brings me back to the question which policy and guidelines are present for quality checking... Also in these cases a bit more information on the actual data would generally be helpful (maybe an issue for the future?).

- I noticed that on some of the individual dataset pages the format of the data is set to "unknown" (for example in the "CIMDEF: a wide-angle deep seismic reflection profile in the Central Iberian Zone" dataset", where all su files are set to "unknown"). It's probably an automatic association/analysis, however, the authors should consider to add this information.

- I could not find index cards for the still embargoed datasets "3D reflection seismic imaging of the Hontomín CO2 storage site" and "SIT4ME: Innovative seismic imaging techniques for mining exploration - Sotiel-Elvira (Spain) dataset". Is this by purpose? Couldn' this type of metadata be made publicly avalable before the end of the embargo? In the text (Line 121) one could get the impression that index cards exist for all datasets.

- L 117: The paper says that there are currently 19 datasets in SeisDARE, however, when I check https://digital.csic.es/handle/10261/101879 I see 26 datsets? Although I understand that SeisDARE is open for new datasets, I think that the paper introducing SeisDARE should be up-to-date and list all of the currently contained ones. Furthermore, I noticed that there are some datsets listed which are not all all DSS dats sets, such as "Regional centroid moment tensors for earthquakes in the 2013 CASTOR gas storage seismic crisis" and "Apatite fission track and zircon (U-Th)/He dataset in the eastern Basque-Cantabrian Zone - western Pyrenees". Please comment. And there are many more... Or is is a mistake? It dos not really fit to the other datasets of SeisDARE... I assumed that the link https://digital.csic.es/handle/10261/101879 is exclusively pointing to SeisDARE?

TECHNICAL CORRECTIONS:

L91: insert hyphen in "continental scale"

L61: insert hyphen in "seismic-related"

L61: What is an "active" data repository? Do you mean voluminous or comprehensive or large data repository?

L141: I do not understand "... being the latter the external foreland zone..." Please check.

L396:insert hyphen in "high-resolution"

———————————————

---

## Referee Comment (RC2) · Florian Haslinger (Referee) · 22 Dec 2020

**GENERAL COMMENTS**

I concur with the observations and support the comments made by Reviewer 1. This paper is a very good introduction of the SeisDARE repository, and the authors are to be congratulated for setting up the repository and introducing it to the community. While the paper includes quite a lot of information about the geological / tectonic settings and some main results of the various experiments, it lacks (also noted by Reviewer 1) in my view some further technical description of the datasets and the criteria applied to the datasets for inclusion in the repository. Thus I support the idea of Reviewer 1 to include a section on technical details / data set descriptions, quality assessment etc. -

and perhaps also add this to the repository website.

**SPECIFIC COMMENTS**

The consistency between the article and the database is now given, the authors apparently removed the non-fitting datasets noted by Reviewer 1. The currently (Dec 2020) 21 datasets listed in https://digital.csic.es/handle/10261/101879 correspond to the paper, as two were added late 2020, after the submission date. If possible, these two datasets could still be added to the paper for the final version (together with any other that might be included by then). As noted above, and in line with the comments from Reviewer 1, an additional section that further describes the technical details of the data sets, any quality check criteria, embargo policies, would be helpful.

**TECHNICAL CORRECTIONS**

In addition to what was noted by Reviewer 1, I just have one additional editorial remark: line 70/71 - the formulation 'The ESFRI in Earth Sciences is the European Plate Observation System...' is a bit misleading, as ESFRI is the Forum under who's guidance / governance ERICs are set up. It would be more correct to write that EPOS is the European Research Infrastructure Consortium for Earth Sciences established under ESFRI. Also note that EPOS now has a new URL https://epos-eu.org

**COMMENTS TO THE REPOSITORY**

From some spot-checks on the repository I noted a couple of items that could be considered (also in addition to what is already noted by Rev 1): - I am not sure what the purpose of the excel file with the 'list of files' is? These 'Description_ficheros_[experiment].xlsx' files just contain the list of files that are part of the dataset (and the filenames don't match the content of the excel file). - quite often the file 'Seismic_data_[experiment]_metadata.xlsx' is missing in the repositories (but is included in the Description_ficheros and Readme... files) - in the file table ('Files in this item') the Description field is usually empty, and the 'Format' field refers to the 'internal'

format of the archive (e.g. a .rar file with trace records has Format entry 'SEGY/su'). While the data file format is an important information, this use of the field is inconsistent and may be confusing (in particular thinking of automatic access & processing). Maybe the internal format could be listed in the description field, and the Format field could specify the actual format of the downloadable file?

---

## Author Comment (AC1) · 18 Jan 2021

Our responses to the reviewer's comments are indicated after each question (in « »).

GENERAL COMMENTS In manuscript essd-2020-208 the authors present a database (SeisDARE) with data of controlled source seismic experiments. Currently, the database contains 19 datasets, most of them covering the Iberian Peninsula and Morocco; it is open for more datsets. The data itself are stored in and accessible at CSIC's GEO3BCN database. Each dataset consists of the metadata and the data in standard format (file-based, i.e. SEG-Y). A DOI is attached to each dataset. The metadata of each data set (including authors/creators, title, characteristics, data examples, references and funding information) is visualized as "landing pages" in the

web-browser and so-called index cards (human readable); it can also be accessed in different (machine-readable) formats. I congratulate the authors for putting together this nice collection of seismic datasets, to preserve them and to make them public for further use following the FAIR idea. The datasets provide a unique opportunity to get insights into the deeper structure of the Iberian Peninsula/Morocco and the processes involved in forming them. Crustal scale seismic profiles are usually very expensive and cannot easily be reproduced. So, keeping and distributing the datasets in this structured way is of high value and is highly interesting for future research (and potentially other purposes). The article is a good way to support the publication of the datasets. The article is logically structured and reads very easy. References for the individual datasets seem complete (see also one of the issues below). I have found only very few typos (see below). The figures are informative and of high quality as is the whole manuscript. A discussion of the accuracy, calibration, processing, etc. are not in detail given in the article, but references have been made to the original articles (in which these aspects should be presented). So I think the topic is extremely well suited to be presented in ESSD I found some issues which I feel the authors should address before the manuscript is eventually ready for publication (moderate revision).

«We really appreciate the interest shown by the reviewer in our manuscript and the recommendation for publication. We also thank the pertinent comments addressed by the reviewer. Therefore, we will resubmit a new version of the manuscript taking into consideration their suggestions, which greatly improve the original version.»

SPECIFIC COMMENTS: - The technical details describing the archived data should be somewhat extended (it seems that the majority of the metadata is to large extend related to the bibliographical data such as authors, title etc.). Actually this is an issue for both the repository and the article. Most of the data (I have checked a few) are stored in SEG-Y format, which is a standard exchange format, however, I suggest to add (in addition to the existing reference/link to SEG) a reference to e.g. Barry, K.M.; Cavers, D.A.; Kneale, C.W. (1975): Recommended standards for digital tape

formats. Geophysics 40 (2): 344– 352). Other datasets are stored in seismic unix (su) format and I think a link or at least a short description or statement regarding this de facto standard format should be added to the article (e.g., https://wiki.seismic-unix.org/doku.php, but there might be better references). In line 120 it says that "Data are mainly in SEG-Y format...". Since datasets in su format is also found, could you be more complete and specific? Is there a policy of the GEO3BCN repository which formats to use? Can you state more specifically which quality control is performed at the GEO3BCN repository? Any checks or guidelines which header words are set or should be provided? Especially for su-files it would be good to know whether it is "old su" format or XDR format, and whether it is little or big endian. I suggest a subsection "Data format" or "Technical issues of the data" in the article which could have at least some general statement regarding these issues (or where I can find this information in the website of the GEO3BCN repository).

«In the new version of the manuscript, a new section has been added: "4 Technical aspects of the data". Here, we briefly discuss the general acquisition parameters of the datasets and the formats of the different files included on the database. References for both SEG-Y and SU formats have been added as suggested. Further information on the details of each dataset can be consulted in the "show full item record" link in the left bottom corner of each dataset and in the publications cited. In addition, part of the information included in section "2 Outline of SeisDARE" has been moved to the current section "4 Technical aspects of the data" for consistency. Additionally, SU files follow the XDR format and have been produced in Linux workstations operating the little endian convention. Using modern versions of SU, the files should be readable in different platforms (Little or big endian, 32 or 64bit). »

- Although I think that su-format is quite handy and widely accepted, it is actually not a standardized exchange format. . . Actually SEG-Y should be preferred...

«SU is an open source and well-stablished format seismic processing package and we have added references to it in the new section 4. Furthermore, in our database only two

up to 21 datasets are exclusively on SU format. Nevertheless, we are working on the conversion of the files from SU to SEG-Y, but this will be completed in the long-term.»

- [more a general comment] I did some spot checks of this large collection of data sets (only spot checks because the data is many hundreds of Gbyte (or even Tbyte); some of the datasets are even embargoed and thus not available). Data conversion and displaying was quite easy e.g. for the IBSERSEIS NI and ILIHA data. Nevertheless, it brings me back to the question which policy and guidelines are present for quality checking... Also in these cases a bit more information on the actual data would generally be helpful (maybe an issue for the future?).

«Assessing the quality of the dataset in a completely objective and numerical way is a difficult issue, and currently we are not aware of a simple procedure. The ESFRI EPOS EU e-infrastructure (https://www.epos-eu.org/) is currently trying to address this issue aiming to define numerical indicators to estimate the quality of geophysical data collections, but this is not yet solved. In the meantime, we prefer to leave out any subjective/qualitative assessment of the data quality to avoid biasing the readers. In any case, each dataset contains a list of publications where the users can observe and estimate the quality of the datasets by themselves.»

- I noticed that on some of the individual dataset pages the format of the data is set to "unknown" (for example in the "CIMDEF: a wide-angle deep seismic reflection profile in the Central Iberian Zone" dataset", where all su files are set to "unknown"). It's probably an automatic association/analysis, however, the authors should consider to add this information.

«We have completed the format field in the repository website, as well as the description field.»

- I could not find index cards for the still embargoed datasets "3D reflection seismic imaging of the Hontomín CO2 storage site" and "SIT4ME: Innovative seismic imaging techniques for mining exploration - Sotiel-Elvira (Spain) dataset". Is this by purpose?

Couldn't this type of metadata be made publicly available before the end of the embargo? In the text (Line 121) one could get the impression that index cards exist for all datasets.

«We would prefer to make the index cards for the embargo datasets available by the time they are released. Nevertheless, it is true that one can get a wrong idea from the text line 121, so we have modified this part of the text to clarify this issue. The new text reads as: "Additionally, an index card summarizes the information for all projects except for those that are embargoed due to data policy."»

- L 117: The paper says that there are currently 19 datasets in SeisDARE, however, when I check https://digital.csic.es/handle/10261/101879 I see 26 datsets? Although I understand that SeisDARE is open for new datasets, I think that the paper instroducing SeisDARE should be up-to-date and list all of the currently contained ones. Furthermore, I noticed that there are some datsets listed which are not all all DSS dats sets, such as "Regional centroid moment tensors for earthquakes in the 2013 CASTOR gas storage seismic crisis" and "Apatite fission track and zircon (U-Th)/He dataset in the eastern Basque-Cantabrian Zone - western Pyrenees". Please comment. And there are many more. . . Or is is a mistake? It does not really fit to the other datasets of SeisDARE. . . I assumed that the link https://digital.csic.es/handle/10261/101879 is exclusively pointing to SeisDARE?

«In the manuscript, we specified that the GEO3BCN database is multidisciplinary and that SeisDARE is part of it, but as it may lead to some confusion, we have re-organized the database website. In the new interface, there is a collection of data that belongs to SeisDARE and another collection with more general type of datasets from GEO3BCN. We hope that with this change it would be easier to follow and understand both the manuscript and the database website. »

TECHNICAL CORRECTIONS: L91: insert hyphen in "continental scale"

«Done»

L61: insert hyphen in "seismic-related" L61: What is an "active" data repository? Do you mean voluminous or comprehensive or large data repository?

«We meant in terms of the volume of data, but we realize that this sentence might be misleading and we have removed it.»

L141: I do not understand ". . . being the latter the external foreland zone..." Please check.

«Within the Iberian Massif only the Cantabrian Zone and the South-Portuguese Zone are external parts of the Variscan orogen. We have clarified it in the text.»

L396: insert hyphen in "high-resolution"

«We have unified the text to "high-resolution".»

---

## Author Comment (AC2) · 18 Jan 2021

The answers to the referee's comments are indicated after each question (in «»).

GENERAL COMMENTS I concur with the observations and support the comments made by Reviewer 1. This paper is a very good introduction of the SeisDARE repository, and the authors are to be congratulated for setting up the repository and introducing it to the community.

«We appreciate the interest shown by the reviewer in our manuscript and thank him for the suitable comments addressed. Therefore, we resubmit a new version of the manuscript taking into consideration his comments. »

While the paper includes quite a lot of information about the geological / tectonic set-

tings and some main results of the various experiments, it lacks (also noted by Reviewer 1) in my view some further technical description of the datasets and the criteria applied to the datasets for inclusion in the repository. Thus I support the idea of Reviewer 1 to include a section on technical details / data set descriptions, quality assessment etc. and perhaps also add this to the repository website.

«In agreement with the comments of Reviewer 1, we have included in the new version of the manuscript a section: "4 Technical aspects of the data". Here, we briefly discuss the general acquisition parameters of the datasets and the formats of the different files included on the database. References for both SEG-Y and SU formats have been added as suggested. We also included the SEG-D format as part of the raw data in the ALCUDIA-NI dataset are in this format. Further information on the details of each dataset can be consulted in the "show full item record" link in the left bottom corner of each dataset and in the publications cited. In addition, part of the information included in section "2 Outline of SeisDARE" has been moved to the current section "4 Technical aspects of the data" for consistency.»

SPECIFIC COMMENTS The consistency between the article and the database is now given, the authors apparently removed the non-fitting datasets noted by Reviewer 1. The currently (Dec 2020) 21 datasets listed in https://digital.csic.es/handle/10261/101879 correspond to the paper, as two were added late 2020, after the submission date. If possible, these two datasets could still be added to the paper for the final version (together with any other that might be included by then).

«Indeed, during the discussion phase we enlarged the list of the datasets in the repository with the IBERSEIS-NI (processed files) and MARCONI-WA. We have consequently included these datasets into the manuscript.»

As noted above, and in line with the comments from Reviewer 1, an additional section that further describes the technical details of the data sets, any quality check criteria,

embargo policies, would be helpful.

«The new section "4 Technical aspects of the data" aims to provide technical information of the data. Regarding the data quality, unfortunately there are no completely objective and quantitative ways to assess this. Currently, EPOS is trying to address this issue aiming to define numerical indicators to estimate the quality of geophysical data collections, but this is not yet solved. In the meantime, we prefer to leave out any subjective/qualitative assessment of the data quality to avoid biasing the readers. In any case, each dataset contains a list of publications where the users can observe and estimate the quality of the datasets by themselves. The embargo comprises a reasonable time period for the use of the data (and potential publication of the main results) by the project members or private companies involved. This has been added to the manuscript for clarity.»

TECHNICAL CORRECTIONS In addition to what was noted by Reviewer 1, I just have one additional editorial remark: line 70/71 - the formulation 'The ESFRI in Earth Sciences is the European Plate Observation System...' is a bit misleading, as ESFRI is the Forum under who's guidance / governance ERICs are set up. It would be more correct to write that EPOS is the European Research Infrastructure Consortium for Earth Sciences established under ESFRI. Also note that EPOS now has a new URL https://epos-eu.org

«In the new version of the manuscript we have re-write that sentence and corrected the link to the EPOS website.»

COMMENTS TO THE REPOSITORY From some spot-checks on the repository I noted a couple of items that could be considered (also in addition to what is already noted by Rev 1): - I am not sure what the purpose of the excel file with the 'list of files' is? These 'Description_ficheros_[experiment].xlsx' files just contain the list of files that are part of the dataset (and the filenames don't match the content of the excel file).

«The 'Description_ficheros_[experiment].xlsx' files included are a requirement of the

repository's file architecture and unfortunately cannot be removed.»

- quite often the file 'Seismic_data_[experiment]_metadata.xlsx' is missing in the repositories (but is included in the Description_ficheros and Readme... files).

«Indeed, as the referee pointed out, there is an inconsistency and we are currently and progressively solving this issue in the repository by updating the 'Description_ficheros_[experiment].xlsx' and 'readme' files. We started creating the file 'Seismic_data_[experiment]_metadata.xlsx' to provide further information on the data and therefore, it was also included into the 'Description_ficheros_[experiment].xlsx'. Finally, as most of the data of the 'Seismic_data_[experiment]_metadata.xlsx' are included along the dataset itself (check also the "show full item record" link), we decided not to upload it. »

- in the file table ('Files in this item') the Description field is usually empty, and the 'Format' field refers to the 'internal' format of the archive (e.g. a .rar file with trace records has Format entry 'SEGY/su'). While the data file format is an important information, this use of the field is inconsistent and may be confusing (in particular thinking of automatic access & processing). Maybe the internal format could be listed in the description field, and the Format field could specify the actual format of the downloadable file?

«The description field in the file table of the repository is being completed with a short and descriptive sentence, and the format field includes the format(s) of the files included.»